# Insight into Steam Permeation through Perovskite Membrane via Transient Modeling

**DOI:** 10.3390/membranes10080164

**Published:** 2020-07-25

**Authors:** Shujuan Zhuang, Ning Han, Qingchuan Zou, Shuguang Zhang, Feng Song

**Affiliations:** 1School of Chemical Engineering, Shandong University of Technology, Zibo 255049, China; en770509@126.com (S.Z.); gregzhangsg@gmail.com (S.Z.); 2Department of Materials Engineering, KU Leuven, 3001 Leuven, Belgium; 3School of Metallurgy, Northeast University, Shenyang 110819, China; zouqingchuan@mail.neu.edu.cn

**Keywords:** separation, perovskite oxide, membrane, steam permeation, modeling

## Abstract

A dynamic model based on BaCe_0.9_Y_0.1_O_3−δ_ (BCY10) perovskite membrane for steam permeation process is presented here to essentially investigate the internal mechanism. The transient concentration distribution and flux of the charged species and the electric potential distribution within the membrane on the steam permeation process are analyzed in detail via simulation based on this model. The results indicate that the flux of steam can be improved via elevating operating temperatures, enlarging the difference of the partial steam pressure between two sides of the membrane, increasing the membrane density, and reducing the membrane thickness. Moreover, it was found that the polarization electric potential between both sides of the membrane occurs during the steam permeation process, especially at the steady state of the steam permeation process. The polarization electric potential reaches the maximum value at about 1050 K in this membrane. The evolution of electric potential can explain the influence of the above-mentioned factors on the steam permeation process. This study advances the mechanism of steam permeation through perovskite membrane, which provides a new strategy for the fundamental investigation of related species permeation (oxygen, carbon dioxide, hydrogen, etc.) on inorganic membranes via transient modeling.

## 1. Introduction

Perovskite oxides with special physical characteristic are widely applied in membrane separation (oxygen/hydrogen/carbon dioxide separation membranes) [1,2,3,4,5,6,7,8,9,10,11,12,13,14,15,16,17,18,19,20,21], fuel cells [22,23,24], and other environment-related application areas [25,26,27,28,29,30,31,32,33,34,35,36,37,38]. Up to now, many studies have focused on the mechanism of perovskite oxide membrane for high-temperature gas separation in practical experiments; however, the research on the simulation of the gas separation process is still limited and inadequate. Since the investigation on the gas separation process would give rise to the insightful understanding of the internal mechanism through the transient data capture [39,40,41], it is urgent to develop and build rational computational simulation model of inorganic membrane materials towards gas separation process.

Among various gas separation processes, it is very important to study the steam permeation process as steam plays a critical role in many devices, particularly solid oxide fuel cells (SOFCs), in which the steam permeation process can increase the flux of proton in the electrolytes. It is worth noting that the charged species permeation during the steam permeation process is closely correlated to the properties of membrane and the operation conditions. Thus, it is necessary and significant to study the relationships between the steam permeation and the above conditions in order to design more efficient membrane reactors.

It is well known that protonic defects would form for certain perovskite materials in water vapor containing environments via the following process.
(1)$$H_{2}O\left(g\right)+V_{O}^{••}+O_{O}^{x}↔2OH_{O}^{•}$$

To be specific, a water molecule enters an oxygen vacancy at the exposed surface of the membrane, donating two protons to the lattice. The quasi-free protons reside near oxygen ions, hopping from lattice site to lattice site by the Grotthuss mechanism. The oxygen ion sub-lattice remains stationary. This reaction can occur at any free surface of the ceramic membrane exposed to water vapor [42]. Oxide ceramic materials with intrinsic oxygen ion vacancies are known to take up and release steam [43], which is greatly meaningful in the steam reforming and SOFC process.

Some previous works have studied the steam permeation process. For example, Coors and his group studied the steam permeation in protonic ceramic fuel cells. In this case, the steady model was built, and the steam permeation flux was predicted [44,45]. In another study, Suksamai and Metcalfe investigated the steam permeation through Y-doped BaCeO_3_ in SOFCs. They found that the steam flux increased from 2 × 10^−4^ to 4 × 10^−4^ mol/m^2^·s when the operation temperature increased from 800 to 1000 K [46].

In this work, we aimed to reveal the mechanism of the steam permeation process on inorganic perovskite oxide membranes via building a dynamic model to simulate the whole process. BaCe_0.9_Y_0.1_O_3−δ_ (BCY10) with perovskite structure was chosen as the target membrane material, and Poisson-Nernst-Planck (PNP) equations were utilized to build the transient model of the mass transfer process of charged species in the solid electrolyte [47,48,49,50]. A transient model on the steam permeation process in the BCY10 membrane was built and the permeation process of proton and oxygen vacancies in BCY10 membrane was simulated via this model. With the PNP equations, the concentration distribution of species in the membrane was calculated, and the evolution curves of the charged species concentration were analyzed. Particularly, the electrical potential between two sides of the membrane during the steam permeation process that is hard to measure by experimental operation was first investigated in detail via this model, providing a new perspective to understand the related species permeation behavior.

## 2. Experimental Section 

### 2.1. Theory

The reaction conditions during the separation process are described below: Wet steam gas is pushed to the surface of the BCY10 membrane, generating a concentration gradient of steam between the gas block phase and the surface of membrane.The steam diffuses through the gas film, and then reacts with oxygen vacancies and oxygen ions, finally leading to the production of protons.On the surface of membrane, the vacancies react with water and the lattice oxygen with proton is produced. Thus, the produced protons diffuse into the membrane and the vacancies diffuse out of the membrane.When the protons permeate through the membrane, the steam is reproduced.

The schematic illustration of the BCY10 membrane for modeling is shown in Figure 1. It is hypothesized that the steam partial pressure on the A side is higher than that on the B side. 

On basis of the above-described steam permeation process, seven steps including gas diffusion, surface reaction, and the charged species permeation processes should be taken into consideration.

The seven steps are as follows:The convection of steam with higher partial pressure in gas phaseThe steam diffusion in the gas boundary layer and Pt layer of Side A of the filmThe reaction of steam with BCY10 on Side A, and the generation of protonsThe migrations of proton and oxygen vacancy within the BCY10 membraneThe reaction of proton, oxygen vacancy, and oxygen ion and the reproduction of steam on Side B of the membraneThe steam diffusion on Side B of the membraneThe convection of the steam with lower partial pressure in gas phase

### 2.2. Mathematical Model 

Since the flat surface of membrane is much larger than the radial surface, the steam permeation process would be treated as one-dimensional process. In this work, the surface concentrations of species (mol/m^2^) are supposed to be equal to the bulk concentrations of species (mol/m^3^) in amount.

Based on the physical model, the mathematics model is the following.

1.The first process can be neglected, due to the negligible effect of gas mass transfer resistance in the bulk gas phase.2.The film diffusion process can be represented with the linear driving force equation and the reaction rate as below:
(2)$$\frac{dC_{S,H_{2}O}\left(t\right)}{dt}=-k_{f}\times \left(C_{H_{2}O}^{*}\left(t\right)-C_{out}\right)-R_{e,H_{2}O}$$
where *k**_f_* is the film diffusion rate constant, m/s; $C_{S,H_{2}O}$ represents the surface concentration of H_2_O, mol/m^2^; $R_{e,H_{2}O}$ means surface reaction rate of H_2_O, mol/(m^2^.s); $C_{H_{2}O}^{*}\left(t\right)$ represents the concentration of steam equilibrium with the concentration of proton in the membrane, mol/m^3^; and *C_out_* refers to the bulk concentration of H_2_O, mol/m^3^.3.The third process represents the water splitting reaction shown below in Equations (3) to (6):
(3)$$H_{2}O\left(g\right)+V_{O}^{••}+O_{O}^{x}\overset{K}{↔}2OH_{O}^{•}$$
(4)$$K=\frac{k_{1}}{k_{2}}=\frac{C_{OH_{O}^{•}}^{2}}{C_{V_{O}^{••}}C_{O_{O}^{x}}P_{H_{2}O}^{*}/P_{0}}$$
(5)$$R_{a}=k_{1}\frac{C_{V_{O}^{••}}C_{O_{O}^{x}}p_{H_{2}o}}{P_{0}}-k_{2}\frac{C_{OH_{O}^{•}}^{2}}{K}$$
(6)$$R_{a}=k_{f}\left(\frac{C_{V_{O}^{••}}C_{O_{O}^{x}}p_{H_{2}o}}{P_{0}}-\frac{C_{OH_{O}^{•}}^{2}}{K}\right)$$
where K is the reaction equilibrium constant of reaction 3, which is calculated with Equation (4); *k*_1_ is the forward reaction rate constant, m^2^/(mol·s); *k*_2_ is the reverse reaction rate constant, m^2^/(mol·s); *R_a_* is the reaction rate of Equation (1), mol/(m^2^·s); and *P*_0_ is the atmospheric pressure, Pa.

The reaction rates of reagents and products can be expressed with Equation (7) to (10), respectively.
(7)$$R_{e,H_{2}O}=-R_{a}$$
(8)$$R_{e,O_{O}^{x}}=-R_{a}$$
(9)$$R_{e,V_{O}^{••}}=-R_{a}$$
(10)$$R_{e,OH_{O}^{•}}=-R_{a}$$
where $R_{e}$ represents the surface reaction rate of species, mol/(m^2^·s).

4.The mass transfer of species, oxygen vacancy ${\mathrm{OH}}_{O}^{•}$ within the membrane can be calculated via the Poisson-Nernst-Planck equation (Equation (11)) [51].
(11)$$\frac{\partial C_{i}\left(x,t\right)}{\partial t}+\frac{\partial }{\partial x}\left(-D_{i}\frac{\partial C_{i}\left(x,t\right)}{\partial x}-z_{i}C_{i}\left(x,t\right)\frac{D_{i}F}{RT}\frac{\partial E\left(x,t\right)}{\partial x}\right)=0$$
where *i* stands for the species permeating in the membrane, *z* stands for the charge number of species, *F* stands for Faraday constant, *R* stands for the ideal gas constant, *T* stands for Kelvin temperature, and *E* stands for the intensity of electric field in the membrane. By Equation (11), the distribution of species in the membrane with time and space can be calculated.

The Poisson equation is used to calculate the intensity of spontaneous electric field in the membrane via Equation (12).
(12)$$\frac{\partial \left[-\varepsilon_{0}\varepsilon_{r}E\left(x,t\right)\right]}{\partial x}=F\overset{N}{\underset{i=1}{\sum }}z_{i}C_{i}\left(x,t\right)$$

5.The steam is reproduced with oxygen vacancy ${\mathrm{OH}}_{O}^{•}$ and $O_{O}^{x}$ in the membrane of BCY10; the rate of reaction can be calculated with Equation (6).6.The diffusion process of steam in the boundary of surface of BCY membrane and the mass transfer rate of steam can be calculated with Equation (2).7.For the electro neutrality of the BCY10 membrane, one global constraint is used, as shown in Equation (13).

(13)$$\overset{N}{\underset{i=1}{\sum }}\int_{0}^{L}z_{i}C_{i}\left(x,t\right)=0$$

Boundary conditions:

For Equation (11), at the boundary of membrane, the inward flux of species is equal to the reaction rate of every species, which is shown as Equations (14)–(17).

On Side A:

X = 0
(14)$$J_{OH_{O}^{•}}=2R_{a}+\frac{dC_{OH_{O}^{•}}}{dt}$$
(15)$$J_{V_{O}^{••}}=-R_{a}+\frac{dC_{V_{O}^{••}}}{dt}$$

On Side B:

X = L
(16)$$J_{OH_{O}^{•}}=2R_{a}+\frac{dC_{OH_{O}^{•}}}{dt}$$
(17)$$J_{V_{O}^{••}}=-R_{a}+\frac{dC_{V_{O}^{••}}}{dt}$$

Here, J stands for the inward flux of species, mol/(m^2^·s).

For Equation (12), on Side A, the electric potential is set as zero, because this side of the membrane is linked to ground, and, on Side B, the electric displacement is set as net charge density in the membrane, which are shown in Equations (18) and (19).

On Side A:

X = 0
(18)$$E\left(x,t\right)=E_{0}$$

On Side B:

X = L
(19)$$-\varepsilon_{0}\varepsilon_{r}E\left(x,t\right)=F\overset{N}{\underset{i=1}{\sum }}z_{i}C_{i}\left(x,t\right)$$

### 2.3. Solution Method 

The partial differential equations in the model were calculated with backward Euler method with the initial condition. The electric field E was calculated with the Poisson equation and the concentrations of charged species were calculated with the Nernst-Planck equation.

## 3. Result and Discussion

### 3.1. Validity of the Model

The reaction equilibrium of the steam permeation based on BCY10 was studied by Kreuer [52] and Coors [44] previously. According to Equation (20), the equilibrium constant depends on the operating temperatures.
(20)$$\ln\left(K\right)=-\frac{\Delta H^{\theta }}{RT}+\frac{\Delta S^{\theta }}{R}$$
where *K* is the reaction equilibrium constant of Equation (1), for BCY10. $\Delta H^{\theta }$ and $\Delta S^{\theta }$ are the enthalpy and entropy of hydration reaction (Equation (1)). R is the gas constant (8.314 J mol^−1^ K^−1^). 

The research results of the enthalpy and entropy of hydration reaction obtained by Kreuer and Coors is shown in Table 1. The enthalpy and the entropy of hydration reaction are negative. According to Equation (20), the reaction equilibrium constant *K_H_* will decrease with the operating temperatures increasing [44]. 

The values of $D_{V_{O}^{••}}$ and $D_{OH_{O}^{•}}$ were determined by Kreuer [43] via Equations (21) and (22). The curves of the diffusion coefficient of proton and oxygen vacancy as a function of the operating temperatures are shown in Figure 2a. It is clear to see that the diffusion coefficient of proton and oxygen vacancy increases with the increase of temperature. It is worth noting that the diffusion coefficient of proton is always larger than that of oxygen vacancy within the set temperature range from 800 to 1200 K.
(21)$$D_{{\mathrm{OH}}_{O}^{•}}=2.03\times 10^{-6}\exp\left(\frac{-52093.55}{RT}\right)$$
(22)$$D_{V_{O}^{••}}=1.10\times 10^{-6}\exp\left(\frac{-69490.29}{RT}\right)$$

Coors studied the steam flux through the BCY10 membrane in the steady state, and the results indicate that the flux of steam increased with the increase of temperature [44]. Metcalfe experimentally measured the steam permeation through the BCY10 membrane, observing that the steam flux increased from 2 × 10^−4^ to 4 × 10^−4^ mol/m^2^·s when the temperature increased from 800 to 1000 K [46].

Moreover, the flux of proton on Sides A and B at different temperatures were calculated via the diffusion coefficients of proton, oxygen vacancies, and the reaction equilibrium coefficient of hydration reaction in BCY10, and the results are displayed in Figure 2b at a given time of 1000 s. The initial conditions used in this study are given in Table 2. As shown in Figure 2b, the flux of proton on both sides increases, and the gap between the two sides becomes smaller with the temperature increasing. This may reflect that the diffusion coefficient of proton increases with the temperature increasing, and the flux of proton at both sides approaches the same value in the steady state of the steam permeation process. According to Equations (7) and (10), the flux of proton is twice that of steam. Thus, it can be concluded that the flux of steam varies from 2 × 10^−4^ to 4 × 10^−4^ mol/m^2^∙s when the temperature increases from 800 to 1000 K. This variation trend of the steam flux with the increase of temperatures agrees with the conclusions drawn from the previous experimental results of Metcalfe and Coors. Thus, the validity of this model has been proved.

### 3.2. The Evolution of the Distribution of Charged Species and the Evolution of the Flux of Proton

To better understand the evolution of flux at different place, we defined “Z” as the differential length across the membrane (0 < Z < L, L = thickness of the membrane). The distribution curves of protons in the membrane at the temperature of 1000 K are shown in Figure 3a. Because both sides of the membrane are exposed to wet gas, the hydration reaction would occur, and the proton concentration would increase at both sides of the membrane. However, on Side A, where the partial pressure of steam is higher, the concentration of proton is higher than that on Side B. This demonstrates that the reaction rate under the higher partial pressure of steam is larger than that under the lower partial pressure of steam. At the initial stage, the concentration of proton on both sides (outside) is larger than that inside the membrane. However, as time goes on, at a certain time, the concentration of proton on Side B is lower than that inside the membrane. This phenomenon can be explained as the reaction (Equation (1)) will occur in reverse and the steam will be reproduced on Side B when protons diffuse through the membrane.

Note that the flux of species is the vector. In this paper, the positive value of flux means the direction of flux from Side A to Side B, while the negative value of flux means the direction of flux from Side B to Side A. The distribution curves of the flux of proton in the membrane at 1000 K are shown in Figure 3b. At the initial stage, the flux of proton is positive on Side A, indicating that the proton diffuses into the membrane from Side A to Side B. In contrast, the flux of proton is negative on Side B, meaning that the proton diffuses into the membrane from Side B to Side A. As the diffusion process proceeds, the flux of proton will decrease on Side A. This may be because the concentration of proton increases inside the membrane, which weakens the mass transfer driving force on Side A. However, on Side B, the absolute value of the flux of proton declines to zero at first and then rises. This may be attributed to the diffusion of protons through the membrane from Side A to Side B, when the steam is reproduced on Side B at a certain time point

### 3.3. The Evolution of the Distribution of the Polarization and the Electric Field in the Membrane

Based on the distributions of species concentration and the flux of proton in the membrane at different temperatures, the polarization in the membrane was also studied. There is no net charge and electron produced in the reaction of hydration. The diffusion flux of species, oxygen vacancy and proton, are not the same as each other. The polarization potential would occur between both sides [48]. The evolution curves of polarization in the membrane at 1000 K are exhibited in Figure 4a. At the beginning stage, the flux of proton is higher at both sides, leading to the strong polarization on both sides. As time goes on, the flux of proton decreases on both sides, resulting in the decrease of the polarization on both sides. Figure 4b displays the evolution of electric field in the membrane. Comparing the curves in Figure 4, it can be found that the evolution tendency of curves at different time are the same, which may demonstrate that the polarization caused by the difference between the flux of the species is the dominant factor to build the electric field in the membrane [53].

### 3.4. The Evolution of the Distribution of the Electric Potential

The distribution curves of electric potential in the membrane at 1000 K are displayed in Figure 5a. The value of potential on Side A is set as zero and all of the electric potentials in the membrane are calculated under this premise. As shown in Figure 5a, at the initial stage, the potential on Side A decreases as the proton diffuses into the membrane. The reaction rate on Side B is smaller than that on Side A, thus the potential on Side B is larger than that on Side A. Subsequently, the difference of potential between both sides increases until the time of 200 s when the potential becomes constant, because the diffusion direction of proton is from Side A to Side B at that time.

The evolution curves of electric potential on Side B at different temperatures are given in Figure 5b. In terms of the temperature rising from 850 to 1000 K, the electric potential first rises and then descends. From 1050 to 1200 K, the electric potential increases monotonically, which may be ascribed to the formation of polarization potential during the diffusion process of species that would react during the diffusion process. That might narrow the difference of species flux until the process of steam permeation reaches the steady state. At the lower temperature range, from 850 to 1000 K, the ratio of diffusion coefficient between proton and oxygen vacancy is approximately 20; meanwhile, the polarization during the species permeation process is strong. Thus, the potential increases abruptly at the beginning, and then the potential decreases slowly as the potential would react upon the charged species during the permeation process. At the higher temperature range, from 1050 to 1200 K, the ratio of diffusion coefficients between proton and oxygen vacancy is around 10. The polarization potential reacts upon the diffusion process mildly, and the electric potential increases slowly. Since the electric potential is the open-circuit voltage between the both sides of the membrane, it can be measured through the voltmeter. However, due to the so high operating temperature, it is almost impossible to measure the electric potential by the voltmeter in the actual experiment. Herein, the simulation of the electric potential during the steam permeation offers a feasible way to get inside into this significant property.

Moreover, the transient electric potential under different temperatures at the time of 1000 s are exhibited in Figure 5c. It is clear to see that the electric potential on Side B increases from 800 to 1050 K and decreases from 1050 to 1200 K. This could be explained by the connection between electric potential and the flux of proton and oxygen vacancy. When the temperature is lower, the diffusion coefficients of oxygen vacancy and proton are smaller, resulting in the smaller electric potential. Once the temperature increases, the reaction equilibrium coefficient of hydration will decrease, which causes the decrease of the equilibrium concentration of proton. Even though both the temperature and the diffusion coefficients of oxygen vacancy and proton increase, the electric potential will decrease at higher temperatures.

### 3.5. The Effects of Thickness and Density of the Membrane on the Flux of Proton

The effect of thickness on the steam permeability was also simulated here via the build-up model. Figure 6a displays the distribution of flux of proton in the membrane with various membrane thicknesses at 1000 s. The flux of proton on Side A is larger than that on Side B, which might be explained as the steam permeation process still not reaching the steady state at 1000 s. The flux of proton decreases with the increase of thickness, which may be attributed to the increase of mass transfer resistance of charged species in the membrane. The evolution curves of flux of proton on Side B of the membrane with different thickness are given in Figure 6b, which indicates that the flux of proton reaches zero earlier for the thinner membrane. As time passes, the proton from Side A permeates through the membrane and the flux of proton on Side B will vary from negative to positive. 

The density of membrane which can be controlled by the preparation technology is another factor to affect the permeability of species. Figure 6d displays the flux of proton in the membrane with different density at 1000 s. The flux of proton is positively correlated with density. The evolution of flux of proton at different density of membrane is shown in Figure 6e. At the beginning, the absolute value of flux of proton on Side B increases with the enhancement of the membrane density, which may be ascribed to the increase of the driving force of mass transfer with the increase of density. At 200 s, the proton from Side A permeates through the membrane, and the flux of proton on Side B increases with the increase of density. Thus, the increase of density gives rise to the increase of flux of proton. 

To better understand the effect of thickness and density from membrane on the flux of proton, we further compared the normalized data in Figure 6c,f. The effect from density of membrane on the flux of proton displays a good linear correlation with the performance (Figure 6f). The constant proportional density changes would bring constant proportional performance changes, while the effect from membrane thickness is not the same (Figure 6c). The increase of thickness will greatly reduce the performance at initial thickening stage and then this effect is diminishing.

### 3.6. The Effects of Partial Pressure of Steam on the Flux of Proton

The partial pressure of steam on both sides of the membrane also has a significant effect on the steam permeation process. In this simulation, the partial pressure of steam on Side A was set as 0.14 atm, while the partial pressure of steam on Side B varied from 0.009 to 0.024 atm. The effect of partial pressure of steam on the flux of proton is given in Figure 7a. The decrease of the partial pressure of steam on Side B induces the increase of the flux of proton. This is ascribed to the improved driving force of mass transfer caused by the increased difference of the partial pressure between Side A and Side B. Figure 7b displays the effect of partial pressure of steam on the evolution of flux of proton on Side B. It is clear that the flux of proton is negative at first, and the absolute value of the proton increases with the increase of the partial pressure of steam on Side B. The time of the flux of proton to reach zero is delayed with increasing the partial pressure of steam, which may be attributed to the enhanced driving force of mass transfer from gas phase to the surface of membrane. We also further calculated the normalized data of the effect from partial pressure of steam on the flux of proton (Figure 7c). It seems that the partial pressure of steam displays a significant linear correlation with the flux of steam.

### 3.7. The Effects of Operating Temperature on the Flux of Proton

The flux of proton at different temperatures at the time of 500 s is exhibited in Figure 8a. As shown in Figure 8a, the flux of proton is larger at higher temperatures. With increasing the temperature, the difference of the flux of proton between both sides is reduced. This might be because the higher temperature improves the diffusion coefficient of proton and promotes the diffusion process to reach the steady state earlier. Figure 8b,c present the evolution curves of the flux of proton at different temperatures on Side A and Side B, respectively. As shown in Figure 8b, with time going on, the evolution curves of flux of proton decreases slowly on Side A, which can be attributed to the decrease of the driving force of mass transfer. The slope of curves reduces with increasing the temperature. This is indicative of the decrease of the hydration reaction equilibrium with the increase of temperatures. At a certain time of 200 s, the flux of proton at the high temperature exceeds that at the low temperature due to the improved diffusion coefficients of proton and oxygen vacancy by increasing temperature. 

As shown in Figure 8c, at the beginning, the flux of proton is negative, which manifests the proton diffuses from Side B to Side A. With the permeation process proceeding, the flux curve increases and the absolute value of flux decreases, which can be attributed to the decrease of the driving force of mass transfer. At a certain time, the flux of proton on Side B becomes positive, which suggests that the proton from Side A has diffused onto Side B. The evolution curves of flux of proton changes from negative to positive and the slop of curves declines with time going on. The diffusion process will reach the steady state, thereby eventually enabling the flux of proton to approach a constant value approach a constant value eventually.

### 3.8. The Validity of the Simulation Resultant

To prove the validity of the simulation presented in this paper, the experiments were operated. The BCY10 membranes were sintered at 1200 °C, and the thickness of membrane was 1.5 mm. The potential difference between the two sides of the membrane was measured at different temperatures (Figure 9a). As shown in Figure 9a, the potential difference increased firstly and then decreased with time going on. Moreover, the potential difference increased and then decreased as temperature rose. The experiment results display a similar trend compared to the simulation results given in Figure 5b. Moreover, the results of steam permeation flux simulation (Figure 2b) also agree well with the experiment (Figure 9b) and the reported work in [54]. Thus, the validity of the simulated results agree well with the experiment. It is a great strategy to bring effective transient modeling in the early stage, which would greatly improve the accuracy of final experiments. 

## 4. Conclusion

One transient model of the steam permeation process based on BCY10 perovskite oxide membrane is proposed, and this model was applied to predict the evolution of the steam permeation process. The polarization potential occurs with the steam permeation process, which is a very important property for exploring the internal mechanism. This model offers a facile way to assess it, as it is hard to measure it via experiment because of the high operating temperatures. The simulation reflects that there is a maximum value of the polarization at the steady state of the steam permeation process. The transient model built in this work would offer some potential value for revealing the internal mechanism of related species permeation (oxygen, carbon dioxide, hydrogen, etc.) on inorganic membranes.

## Figures and Tables

**Figure 1 membranes-10-00164-f001:**
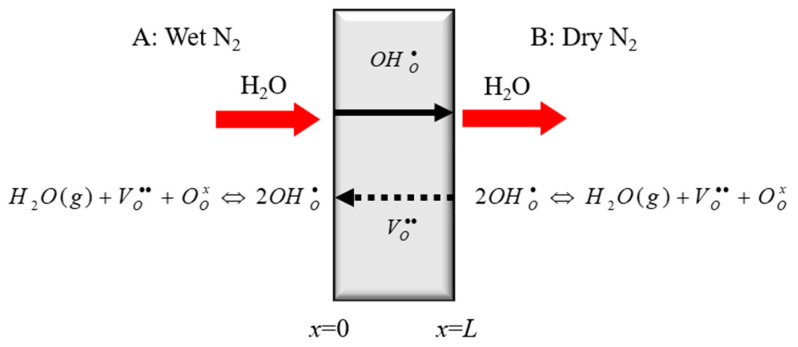
Schematic illustration of BCY10 membrane for modeling (L = membrane thickness).

**Figure 2 membranes-10-00164-f002:**
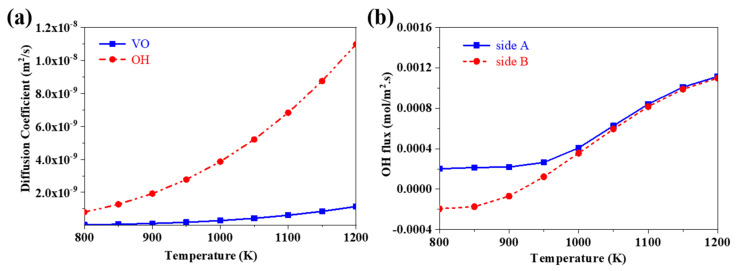
(**a**) The effect of temperature on the diffusion coefficient of proton (OH) and oxygen vacancy (VO); and (**b**) the flux of proton on Sides A and B at different temperatures.

**Figure 3 membranes-10-00164-f003:**
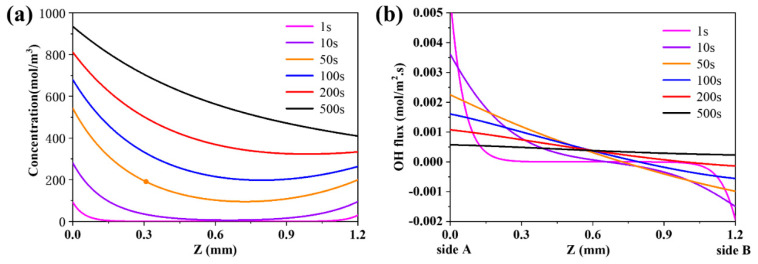
(**a**) The evolution of the distribution of proton concentration in the BCY10 membrane; and (**b**) the evolution of the flux of proton at 1000 K.

**Figure 4 membranes-10-00164-f004:**
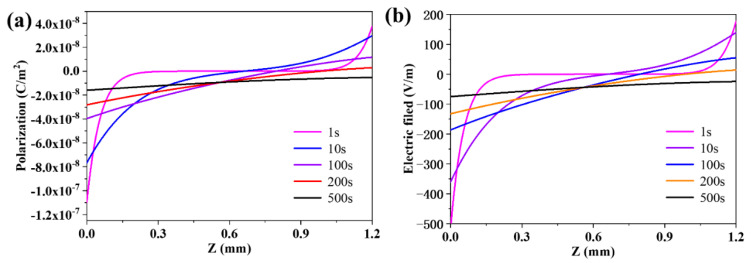
(**a**) The evolution of polarization in the membrane; and (**b**) the evolution of electric field in the membrane.

**Figure 5 membranes-10-00164-f005:**
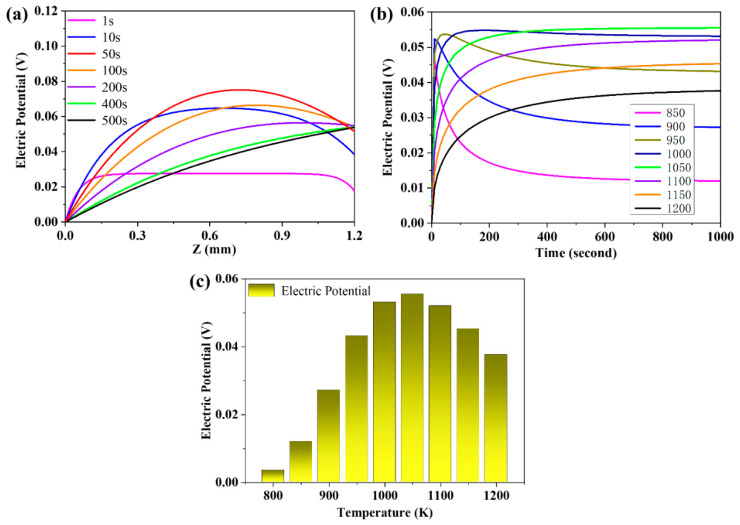
(**a**) The distribution of electric potential in the membrane at 1000 K; (**b**) the evolution curves of electric potential on Side B at different temperatures; and (**c**) the electric potential on Side B at the time of 1000 s.

**Figure 6 membranes-10-00164-f006:**
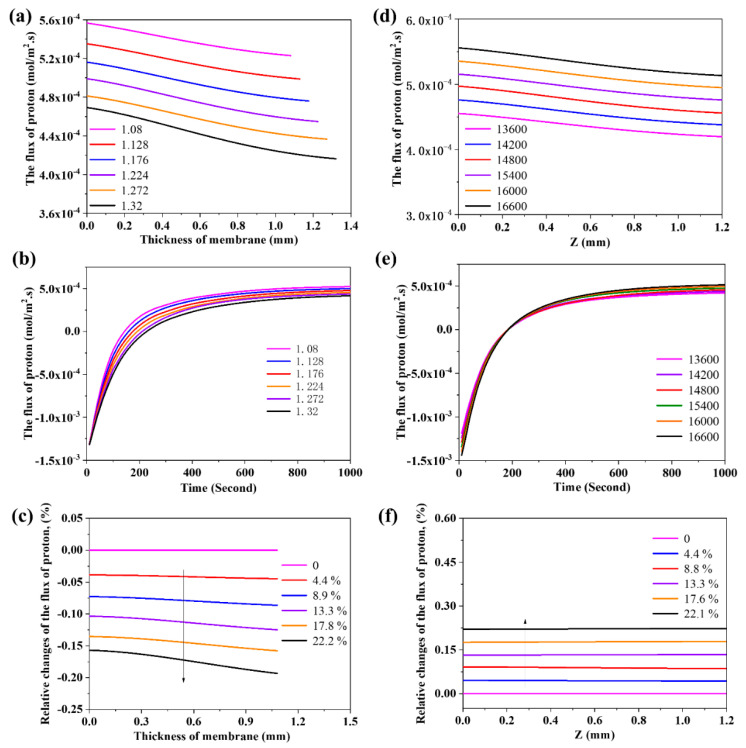
(**a**) The effect of thickness of membrane on the flux of proton; (**b**) the effect of thickness of membrane on the evolution of flux of proton; (**c**) normalized analysis of the effect from thickness of membrane on the flux of proton; (**d**) The effect of density of membrane on the flux of proton; (**e**) the effect of density of membrane on the evolution of flux of proton; and (**f**) normalized analysis of the effect from density of membrane on the flux of proton.

**Figure 7 membranes-10-00164-f007:**
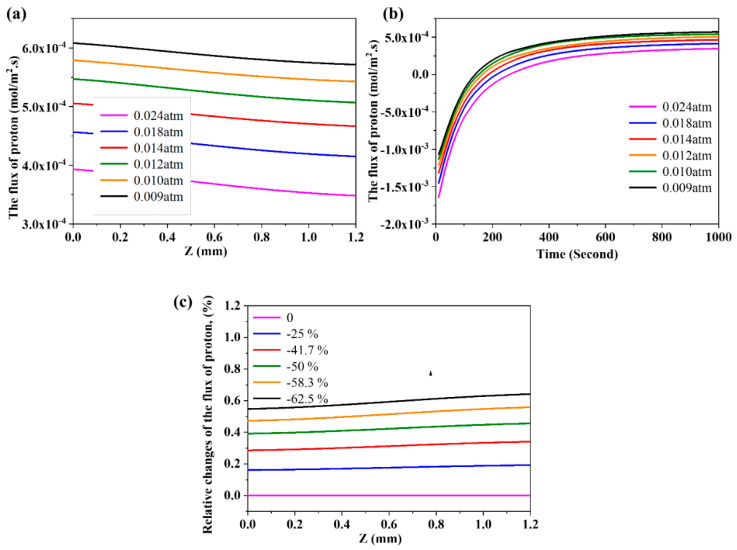
(**a**) The effect of partial pressure of steam on the flux of proton; (**b**) the effect of partial pressure of steam on the evolution of flux of proton; and (**c**) normalized analysis of the effect from partial pressure of steam on the flux of proton.

**Figure 8 membranes-10-00164-f008:**
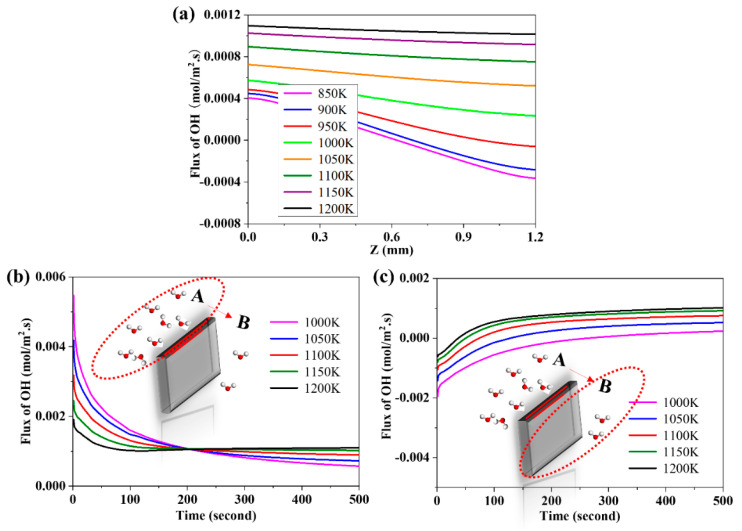
(**a**) The effect of temperature on the flux of proton; (**b**) the evolution of the flux of proton at Side A; and (**c**) the evolution of the flux of proton at Side B.

**Figure 9 membranes-10-00164-f009:**
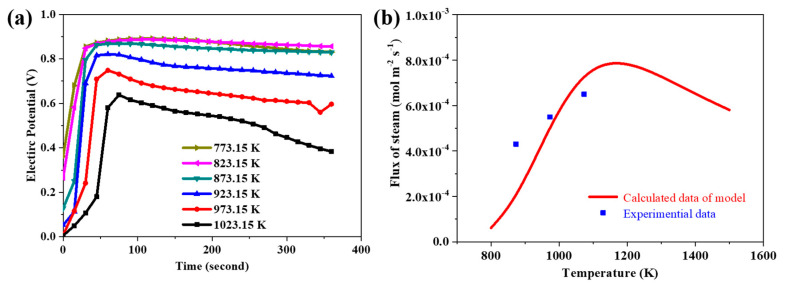
(**a**) The curves of the difference of electric potential between both sides as a function of time at difference temperatures; and (**b**) the steam flux variation through the membrane at different temperatures.

**Table 1 membranes-10-00164-t001:** Hydration enthalpy and entropy at a constant steam partial pressure of 0.025 atm.

Material	$\Delta H^{\theta }(\mathrm{KJ}/\mathrm{mol})$	$\Delta S^{\theta }(J/\mathrm{mol}\cdot K)$	Investigator
BCY10	−162.2	−166.7	Kreuer
BCY10	−156.1	−145.2	Coors

**Table 2 membranes-10-00164-t002:** The initial parameters of the model in this work.

Experimental Parameters	Values
Concentration of oxygen ions	4.4633 × 10^4^ mol/m^3^
Concentration of Y doped Ce	1.513 × 10^3^ mol/m^3^
Concentration of oxygen vacancy	7.565 × 10^2^ mol/m^3^
Concentration of Proton	0
partial pressure of steam	Side A 0.14 atm; Side B 0.014 atm
V (the initial electric potential)	0
k1 (mass transfer coefficient of boundary layer)	30 m/s
L (thickness of membrane)	1.2 mm
pullet radius	7 mm

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
