# Peer review of "Insight into Steam Permeation through Perovskite Membrane via Transient Modeling"

_membranes, 2020, doi:10.3390/membranes10080164_

Round 1

Reviewer 1 Report

The authors submitted the work titled “Insight into steam permeation through perovskite membrane via transient modeling” The authors need to answer some concerns about the modeling results before considering this work for publication. Also, some parts of the introduction, the mathematical model explanation, the validation of the model and some results as the permeation flux of steam (OH flux as authors refer) vs. temperature, electric potential vs. temperature, and permeation flux of steam vs. thickness were presented and described in their previous work (Feng Song, Shujuan Zhuang, Xiaoyao Tan, Shaomin Liu, Modeling of Steam Permeation through the High-Temperature Proton Conducting Ceramic Membranes, AIChE Journal, 2018, 65, 777-782, DOI 10.1002/aic.16468). So, it is highly recommended that they cite previous work related to this study, and only present the novelty of new insights in this work.

Ce-based materials in non-oxidative atmospheres are reduced from Ce4+ to Ce3+. So, why was it not considered the electronic conductivity in the modeling if this affects the dissociation of H2O?

Moderate English changes and uses of appropriate technical language are required. For example, in some cases, they use “OH flux” (Figure 3) and, in other cases, “The flux of proton” (Figure 6). Both terminologies can also be confused with the Permeation flux of H2O (JH2O). Is JH2O what it is presented in figure 3? From figures 3 to 8 it was used “Thickness” as the label of the x-axis. However, this terminology can be confused with the actual thickness of the membrane. It seems like it was varied the thickness of the membrane. It is recommended to use other variables, for example, “z” as differential length across the membrane (0<z<L, L=thickness of the membrane).

Line 83, what do “vacancies diffuse out of the membrane” mean? In what part of the H2O dissociation the vacancies go out of the membrane?

It could be possible to double-check eq. 7 and 10? In the feed side of the membrane (side A) OH is a product, so should the reaction rate of OH be positive? Actually, in line 201, it is mentioned, “According to Eqn.7and Eqn.10, the flux of proton is twice that of steam.” Nevertheless, it does not seem to be accurate or precise.

Driving force for OH diffusion is related to the partial pressure gradient of steam across the membrane. Fig 3a shows that OH concentration in A is higher than in B.  However, during the first 200sec there is a permeation flux from B to A. It is still not clear why at some length across the membrane the OH concentration is lower than in site B. What is happening with OH, why the resistance is higher at this point?

Line 286-283, it is mentioned “Even though both the temperature and the diffusion coefficients of oxygen vacancy and proton increases, the electric potential will decrease at a higher temperature” and “Once the temperature increases, the reaction equilibrium coefficient of hydration will decrease, which causes the decrease of the equilibrium concentration of proton.” However, Fig 2b shows that the proton concentration increases with temperature. So, what would be the explanation that the electric potential decreases at a higher temperature?

Author Response

Responses to Reviewer #1

Original comment of Reviewer #1: The authors submitted the work titled “Insight into steam permeation through perovskite membrane via transient modeling” The authors need to answer some concerns about the modeling results before considering this work for publication.

Reply and revisions: We thank Reviewer#1 for his/her positive evaluation and time contribution. Base on your suggestions, we have tried our best to revise it in this hard time. Thanks again for your kind understanding and precious time on our case. We have revised the comments one by one as below:

Q1. Reviewer #1: Some parts of the introduction, the mathematical model explanation, the validation of the model and some results as the permeation flux of steam (OH flux as authors refer) vs. temperature, electric potential vs. temperature, and permeation flux of steam vs. thickness were presented and described in their previous work (Feng Song, Shujuan Zhuang, Xiaoyao Tan, Shaomin Liu, Modeling of Steam Permeation through the High-Temperature Proton Conducting Ceramic Membranes, AIChE Journal, 2018, 65, 777-782, DOI 10.1002/aic.16468). So, it is highly recommended that they cite previous work related to this study, and only present the novelty of new insights in this work.

Reply and revisions: We thank Reviewer#1’s good suggestion.

As you suggested, we have cited this work in suitable place. Our previous work published in AICHE (AIChE Journal, 2018, 65, 777-782, DOI 10.1002/aic.16468) is focused on affection of operation parameters on the steam flux of membrane. However, in this paper, the distribution of electric potential and charged species are focused. Thus, the discussions in this paper are deeper on the permeation theory of charged species in the membrane.

Q2. Reviewer #1: Ce-based materials in non-oxidative atmospheres are reduced from Ce4+ to Ce3+. So, why was it not considered the electronic conductivity in the modeling if this affects the dissociation of H2O?

Reply and revisions: We thank Reviewer#1’s good suggestion. In the lower steam pressure conditions, the reduced of Ce is very less, and the experiment time is not long enough. So that, the changes on oxidation of Ce is ignore. We would like to further detailly investigated this phenomenon. 

Q3. Reviewer #1: Moderate English changes and uses of appropriate technical language are required. For example, in some cases, they use “OH flux” (Figure 3) and, in other cases, “The flux of proton” (Figure 6). Both terminologies can also be confused with the Permeation flux of H2O (JH2O). Is JH2O what it is presented in figure 3? From figures 3 to 8 it was used “Thickness” as the label of the x-axis. However, this terminology can be confused with the actual thickness of the membrane. It seems like it was varied the thickness of the membrane. It is recommended to use other variables, for example, “z” as differential length across the membrane (0<z<L, L=thickness of the membrane).

Reply and revisions: We thank Reviewer#1’s good suggestion. We have revised all the figures as you suggested.

Q4. Reviewer #1: Line 83, what do “vacancies diffuse out of the membrane” mean? In what part of the H2O dissociation the vacancies go out of the membrane?

Reply and revisions: Thanks for your suggestion. We have revised as below:

“On the surface of membrane, the vacancies will react with water and the lattice oxygen with proton is produced. Thus, the produced protons will diffuse into the membrane and the vacancies diffuse out of the membrane.”

Q5. Reviewer #1: It could be possible to double-check eq. 7 and 10? In the feed side of the membrane (side A) OH is a product, so should the reaction rate of OH be positive? Actually, in line 201, it is mentioned, “According to Eqn.7and Eqn.10, the flux of proton is twice that of steam.” Nevertheless, it does not seem to be accurate or precise.

Reply and revisions: Thanks for your suggestion and carefulness.

The steam permeation process in the BCY membrane can be considered as two parties. In the beginning of the process, the concentration of species and the electric potential in the membrane changed sharply, and with time increase, the flux of species and the electric potential became the steady state. At the steady state, the flux of OH is twice that of steam. However, at the beginning state, the equation is not right.

Q6. Reviewer #1: Driving force for OH diffusion is related to the partial pressure gradient of steam across the membrane. Fig 3a shows that OH concentration in A is higher than in B. However, during the first 200sec there is a permeation flux from B to A. It is still not clear why at some length across the membrane the OH concentration is lower than in site B. What is happening with OH, why the resistance is higher at this point?

Reply and revisions: As shown in the table, 2, the partial pressure of steam at A is larger than that at B. At the beginning of steam permeation process, the steam at both sides should react with BCY membrane and the OH permeates into the membrane. However, the concentration of OH from side A is larger than that from side B. So, at a certain time, the reverse reaction should happen at side B, then the steam is produced.

Q7. Reviewer #1: Line 286-283, it is mentioned “Even though both the temperature and the diffusion coefficients of oxygen vacancy and proton increases, the electric potential will decrease at a higher temperature” and “Once the temperature increases, the reaction equilibrium coefficient of hydration will decrease, which causes the decrease of the equilibrium concentration of proton.” However, Fig 2b shows that the proton concentration increases with temperature. So, what would be the explanation that the electric potential decreases at a higher temperature?

Reply and revisions: We have done the simulation of steam permeation (Figure 9), which indicates that this tendency would appear at rather high temperature range. But the temperature is not large enough to show the same curve.

Reviewer 2 Report

The manuscript can only be accepted for publication in Membranes after a major revision of the text. Detailed comments are given below.

In order to validate the simulation results based on this model, please offer some experimental data.

Please give a detail and good mechanism figure of the steam permeation process on inorganic perovskite oxide membranes. 

Line 111-112 and line 121, and line 128-129, and line 139 and line 171、179: Can these physical quantity representations be replaced by the Formula editor?

Line 131: What did the E stand for? Is it stands for the potential field?

Line 149-153: Please indicate the “J”.

Line 170-171: Please unify the KH and ln(K) in the formula(20) and explanation.

Can the phrase“the polarization caused by the difference between the flux of the species is the dominant factor to build the electric filed in the membrane. ”be supported by data from the literature?

Author Response

Responses to Reviewer #2

Original comment of Reviewer #2: The manuscript can only be accepted for publication in Membranes after a major revision of the text. Detailed comments are given below.

Reply and revisions: We thank for Reviewer#2’s comments and time contribution on our work. Base on your suggestions, we have tried our best to revise it in this hard time. Thanks again for your kind understanding and precious time on our case. We have revised the comments one by one as below:

Q1. Reviewer #2: In order to validate the simulation results based on this model, please offer some experimental data.

Reply and revisions: We thank Reviewer#2’s good suggestion. We have added some experiment resultant and discussion into manuscript.

Figure 9. (a) The curves of the difference of electric potential between both sides as a function of time at difference temperatures; (b) The steam flux variation through the membrane at different temperatures.

3.8 The validity of the simulation resultant

To prove the validity of simulation in this paper, the experiments were operated. The BCY10 membranes are sintered at 1200 °C, the thickness of membrane is 1.5mm. The potential difference between the two sides of the membrane was measured at different temperatures (Figure 9a). As shown in Figure 9a, the potential difference increases firstly and then decreases with time going on. Moreover, the potential difference increases and then decreases as temperature rising. The experiment resultant displays the similar trend compared to the simulation resultant given in Figure 5b. Moreover, the resultant of steam permeation flux simulated (Figure 2b) also agrees well with the experiment (Figure 9b) and the reported work [54]. Thus, the validity of the simulated resultant agrees well with experiment. It’s a great strategy to bring effective transient modeling in the early stage, which would greatly improve the accuracy of final experiment.

Q2. Reviewer #2: Please give a detail and good mechanism figure of the steam permeation process on inorganic perovskite oxide membranes.

Reply and revisions: We thank Reviewer#2’s good suggestion. We have revised this in the manuscript.

Figure 1. Schematic illustration of BCY10 membrane for modeling (L= membrane thickness)

Q3. Reviewer #2: Line 111-112 and line 121, and line 128-129, and line 139 and line 171、179: Can these physical quantity representations be replaced by the Formula editor?

Reply and revisions: Thanks for Reviewer#2’s suggestion. We have revised this part in the manuscript as below:

Where kf is the film diffusion rate constant, m/s;  represents the surface concentration of H2O, mol/m2;  means surface reaction rate of H2O, mol/(m2.s),  represents the concentration of steam equilibrium with the concentration of proton in the membrane, mol/m3; Cout refers to the bulk concentration of H2O, mol/ m3.

Q4. Reviewer #2: Line 131: What did the E stand for? Is it stands for the potential field?

Reply and revisions: Thanks for Reviewer#2’s suggestion. We have revised this part in the manuscript as below:

“E stands for the intensity of electric field in the membrane.”

Q5. Reviewer #2: Line 149-153: Please indicate the “J”.

Reply and revisions: Thanks for Reviewer#2’s suggestion. We have revised this part in the manuscript as below:

“Here, J stands for the inward flux of species, mol/(m2·s).”

Q6. Reviewer #2: Line 170-171: Please unify the KH and ln(K) in the formula(20) and explanation.

Reply and revisions: Thanks for Reviewer#2’s suggestion. We have revised this part in the manuscript as below:

“Where K is the reaction equilibrium constant of Eqn.1, for BCY10.  and  are the enthalpy and entropy of hydration reaction (Eqn 1). R is the gas constant, (8.314 J mol−1 K−1). ”

Table 1. Hydration enthalpy and entropy at a constant steam partial pressure of 0.025 atm

Material

(KJ/mol)

(J/mol·K)

Investigator

BCY10

-162.2

-166.7

Kreuer

BCY10

-156.1

-145.2

Coors

Q7. Reviewer #2: Can the phrase “the polarization caused by the difference between the flux of the species is the dominant factor to build the electric filed in the membrane. ” be supported by data from the literature?

Reply and revisions: Thanks for Reviewer#2’s suggestion. The phrase which means that the difference between the flux of the species should cause the difference concentrations of the species, the difference concentration of the species can cause the concentration polarization. As you suggested, we have cited one reference here with the same phenomenon.

Deng, L. , Feng, X. , Ren, G. K. , Wang, J. , & Song, J. . (2019). Effects of structural and concentration polarization on the efficiency of pd membrane permeator. Fusion Engineering and Design, 148, 111279.

Reviewer 3 Report

This article presents the results of the dynamic model based on BaCe0.9Y0.1O3-δ (BCY10) perovskite membrane for steam permeation process. The model takes into account the effect of temperature on the diffusion coefficient of proton and oxygen vacancy, the flux of proton on the side A and B at different temperatures, the distribution of proton concentration in the membrane, the distribution of electric potential in the membrane, the effect of temperature, thickness and density of membrane and the partial pressure of steam on the flux of proton. Although the results are important and useful, their presentation is of little interest. I would suggest changing the arrangement of the manuscript to present the most important things resulting from the model used. To this purpose, I suggest the following actions:

  • grouping together the results for the properties of the membrane (thickness and density) as well as the results for the process parameters.
  • comparison of the importance of individual parameters for the transport of protons through the membrane.
  • analyse the model on specific measurement data for several membranes and prove that the best membrane can be selected from the proposed model.

I recommended to publish this work after major revisions.

Author Response

Responses to Reviewer #3

Original comment of Reviewer #3: This article presents the results of the dynamic model based on BaCe0.9Y0.1O3-δ (BCY10) perovskite membrane for steam permeation process. The model takes into account the effect of temperature on the diffusion coefficient of proton and oxygen vacancy, the flux of proton on the side A and B at different temperatures, the distribution of proton concentration in the membrane, the distribution of electric potential in the membrane, the effect of temperature, thickness and density of membrane and the partial pressure of steam on the flux of proton. Although the results are important and useful, their presentation is of little interest. I would suggest changing the arrangement of the manuscript to present the most important things resulting from the model used. To this purpose, I suggest the following actions:

Reply and revisions: We thank for Reviewer#3’s comments and time contribution on our work. Base on your suggestions, we have tried our best to revise it in this hard time. Thanks again for your kind understanding and precious time on our case. We have revised the comments one by one as below:

Some comments:

Q1. Reviewer #3: Grouping together the results for the properties of the membrane (thickness and density) as well as the results for the process parameters.

Reply and revisions: We thank for Reviewer#3 for his/her good suggestion. We have further added Figure 6c and 6f of the normalized analysis of the effect from thickness and density of membrane on the flux of proton. And also add some sentence to discussed these figures in the manuscript.

“To better understand the effect from thickness and density from membrane on the flux of proton. We further compared the normalized data in Figure 6c&f. The effect from density of membrane on the flux of proton displays a good linear correlation with the performance (Figure 6f). The constant proportional density changes would bring constant proportional performance changes. While the effect from membrane thickness is not the same (Figure 6c). The increase of thickness will greatly reduce the performance at initial thickening stage and then this effect is diminishing.”

Figure 6. (a) The effect of thickness of membrane on the flux of proton; (b) the effect of thickness of membrane on the evolution of flux of proton; (c) normalized analysis of the effect from thickness of membrane on the flux of proton; (d) The effect of density of membrane on the flux of proton; (e) the effect of density of membrane on the evolution of flux of proton; (f) normalized analysis of the effect from density of membrane on the flux of proton.

Q2. Reviewer #3: Comparison of the importance of individual parameters for the transport of protons through the membrane.

Reply and revisions: We thank for Reviewer#3 for his/her good suggestion. We have also further added Figure 7c of the normalized analysis of the effect from partial pressure of steam on the flux of proton. And also add some sentence to discussed these figures in the manuscript.

Figure 7. (a) The effect of partial pressure of steam on the flux of proton; (b) the effect of partial pressure of steam on the evolution of flux of proton; (c) normalized analysis of the effect from partial pressure of steam on the flux of proton.

Q3. Reviewer #3: Analyse the model on specific measurement data for several membranes and prove that the best membrane can be selected from the proposed model.

Reply and revisions: Thanks for Reviewer#3 for his/her good suggestion. This work is focus on BaCe0.9Y0.1O3-δ (BCY10) perovskite membrane, as the structure parameters are according to this material. You have raised good suggestions, and we could further try to investigate various materials through this model to select the best membrane. 

Finally. Reviewer #3: I recommended to publish this work after major revisions.

Reply and revisions: We thank Reviewer#3 for his/her positive evaluation and time contribution.

Round 2

Reviewer 1 Report

The authors have improved the manuscript titled: “Insight into steam permeation through perovskite membrane via transient modeling.” The authors answered all the questions made in the first round revision.

After the following minor revisions, It is recommended to accept this study:

1) The X-axis in Figure 6A and 6C need to be changed by “Z (mm)”

2) The Y-axis in Figure 9B needs to be replaced by “Permeation flux of steam.”

3) The Y-axis in Figure 7 and 6 need to be replaced by “Flux of protons.”

Author Response

Responses to Reviewer #1

Original comment of Reviewer #1: The authors have improved the manuscript titled: “Insight into steam permeation through perovskite membrane via transient modeling.” The authors answered all the questions made in the first-round revision. After the following minor revisions, It is recommended to accept this study:

Reply and revisions: We thank Reviewer#1 for his/her positive evaluation and time contribution to this work. We have revised the comments one by one as below:

Q1. Reviewer #1: The X-axis in Figure 6A and 6C need to be changed by “Z (mm)”

Reply and revisions: We thank Reviewer#1’s carefulness. I have further revised this Figure 6.

Q2. Reviewer #1: The Y-axis in Figure 9B needs to be replaced by “Permeation flux of steam.”

Reply and revisions: We thank Reviewer#1’s good suggestion. We have revised this in Figure 9.

Q3. Reviewer #1: The Y-axis in Figure 7 and 6 need to be replaced by “Flux of protons.”

Reply and revisions: We thank Reviewer#1’s good suggestion. We have revised the Figure 6&7 as you suggested.

Reviewer 2 Report

The manuscripts have been improved very much after review. I suggest that it is pulished in membrances. but have a little  mistake to correct in below.

  1 Please unify significant digits in manuscripts.

  2. Tables should be three line tables.

Author Response

Responses to Reviewer #2

Original comment of Reviewer #2: The manuscripts have been improved very much after review. I suggest that it is published in membranes. but have a little mistake to correct in below.

Reply and revisions: We thank for Reviewer#2’s comments and time contribution on our work. We have revised the comments one by one as below:

Q1. Reviewer #2: Please unify significant digits in manuscripts.

Reply and revisions: We thank Reviewer#2’s good suggestion. We have double checked the manuscript.

Q2. Reviewer #2: Tables should be three-line tables.

Reply and revisions: We thank Reviewer#2’s good suggestion. We offer the word document not only PDF. The editors would make the final formatting according to this journal, once this work has been accepted. Thanks for your kind understanding.

Reviewer 3 Report

Presented revised article shows the results of the dynamic model based on BaCe0.9Y0.1O3-δ (BCY10) perovskite membrane for steam permeation process. The authors took into account the reviewer's comments and added the relevant text, tables and figures to the manuscript. I suggest to accept the article after changing the sentence “To better understand the effect from thickness and density from membrane on the flux of proton” (line 313) because this sentence is not written correctly.

Author Response

Responses to Reviewer #3

Original comment of Reviewer #3: Presented revised article shows the results of the dynamic model based on BaCe0.9Y0.1O3-δ (BCY10) perovskite membrane for steam permeation process. The authors took into account the reviewer's comments and added the relevant text, tables and figures to the manuscript. I suggest to accept the article after changing the sentence “To better understand the effect from thickness and density from membrane on the flux of proton” (line 313) because this sentence is not written correctly.

Reply and revisions: We thank for Reviewer#3’s comments and time contribution on our work. Base on your suggestions, we have tried our best to revise it. Thanks again for your kind understanding and precious time on our case.

We have revised this sentence asTo better understand effects of the thickness and density of the membrane on the flux of protons
